# Higher-order olfactory neurons in the lateral horn support odor valence and odor identity coding in *Drosophila*

**Sudeshna Das Chakraborty[1], Hetan Chang[2], Bill S Hansson[2†], Silke Sachse[1\*†]**

[1]Research Group Olfactory Coding, Max Planck Institute for Chemical Ecology, Jena, Germany; [2]Department of Evolutionary Neuroethology, Max Planck Institute for Chemical Ecology, Jena, Germany

**Abstract** Understanding neuronal representations of odor-evoked activities and their progressive transformation from the sensory level to higher brain centers features one of the major aims in olfactory neuroscience. Here, we investigated how odor information is transformed and represented in higher-order neurons of the lateral horn, one of the higher olfactory centers implicated in determining innate behavior, using *Drosophila melanogaster*. We focused on a subset of third-order glutamatergic lateral horn neurons (LHNs) and characterized their odor coding properties in relation to their presynaptic partner neurons, the projection neurons (PNs) by two-photon functional imaging. We show that odors evoke reproducible, stereotypic, and odor-specific response patterns in LHNs. Notably, odor-evoked responses in these neurons are valence-specific in a way that their response amplitude is positively correlated with innate odor preferences. We postulate that this valence-specific activity is the result of integrating inputs from multiple olfactory channels through second-order neurons. GRASP and micro-lesioning experiments provide evidence that glutamatergic LHNs obtain their major excitatory input from uniglomerular PNs, while they receive an odor-specific inhibition through inhibitory multiglomerular PNs. In summary, our study indicates that odor representations in glutamatergic LHNs encode hedonic valence and odor identity and primarily retain the odor coding properties of second-order neurons.

**\*For correspondence:**
ssachse@ice.mpg.de

†These authors share senior authorship

**Competing interest:** The authors declare that no competing interests exist.

## Editor's evaluation

Information about the environment, obtained through sensory organs, is processed and utilized at multiple levels in the brain. In this study, the authors use a variety of modern genetic and optophysiological tools to uncover the function and connectivity of glutamatergic neurons in a higher brain center of *Drosophila* – the lateral horn. They find that these neurons do not only encode chemical odor identity, but also the hedonic value (attractive or repulsive) of odors. This advances our understanding of how odors are represented in the brain and will be of value to those who are interested in odor coding and behavioral valence of various odors.

## Introduction

Insects are the most successful taxon among the whole animal kingdom in terms of their distribution and ability to survive in a multitude of environmental conditions. Largely they rely on their olfactory sense to carry out their fundamental goal-directed behaviors, such as food navigation, mating, ovipositing, or escape from predators. The powerful ability to detect odor cues, to evaluate the information efficiently with a relatively small number of neurons and to transform the neuronal signal into an appropriate behavioral output, makes the insect olfactory system a premier model system

for olfactory research. Numerous studies have investigated the neuronal representation of odors at successive neuronal layers from the periphery to higher brain levels using *Drosophila melanogaster* as a model organism (*Bhandawat et al., 2007*; *Ng, 2002*; *Root et al., 2007*; *Schmuker et al., 2007a*; *Schmuker and Schneider, 2007b*; *Seki et al., 2017*; *Wilson et al., 2004*) Although much progress has been made in understanding odor coding at the antennal lobe (AL) level (*Bhandawat et al., 2007*; *Galizia, 2014*; *Ng, 2002*; *Wilson et al., 2004*), the coding strategies and processing mechanisms of higher brain centers still remain largely elusive. In this regard, the lateral horn (LH) has recently gained attention as a crucial signal processing center integrating both innate and learned behavioral information (*Das Chakraborty and Sachse, 2021*). Several studies during recent years have advanced our understanding of the anatomical and functional properties of higher-order lateral horn neurons (LHNs) regarding odor processing (*Das Chakraborty and Sachse, 2021*; *Dolan et al., 2019*; *Frechter et al., 2019*; *Jeanne et al., 2018*; *Lerner et al., 2020*; *Varela et al., 2019*). The generation of several LH cell-type-specific lines, characterization of polarity and neurotransmitter identity of LHNs, as well as the establishment of detailed EM connectomic datasets have led to a significant progress in the field to study the function of specific LHN classes (*Dolan et al., 2019*; *Frechter et al., 2019*). The LH is comprised of three categories of neurons, which include LH input neurons (LHINs, which are mainly olfactory projection neurons [PNs] along with mechanosensory, thermosensory, and gustatory neurons), LH local neurons (LHLNs), and LH output neurons (LHONs) (*Bates et al., 2020*; *Dolan et al., 2019*; *Frechter et al., 2019*). In terms of PN-LHN connectivity, the olfactory PNs deriving from individual glomeruli of the AL form stereotyped and conserved connections with certain LHNs (*Fişek and Wilson, 2014*; *Jeanne et al., 2018*; *Jefferis et al., 2007*; *Marin et al., 2002*; *Wong et al., 2002*). Although all kinds of connections are possible, PNs having similar odor-tuning patterns are prone to target similar LHN types (*Jeanne et al., 2018*). Certain pairs of narrowly tuned glomeruli encoding ecologically relevant odors and eliciting specific kinds of behavior, such as courtship, aggregation, or food seeking, converge onto the same LHN types and have been shown to be overrepresented in the LH in terms of synaptic densities (*Bates et al., 2020*; *Jeanne et al., 2018*). Furthermore, a high amount of divergence has also been described to occur at the level of PN to LHN connectivity (*Huoviala et al., 2018*). Altogether, these complex connectivity patterns in addition to direct pooling of feedforward inputs from PNs innervating different glomeruli result in broader tuning patterns of LHNs compared to their presynaptic PNs (*Bates et al., 2020*; *Frechter et al., 2019*). In addition to the observed broadly tuned LHNs, narrowly tuned LHNs also exist, which receive input from a single type of PN and which are assumed to be further modulated by odorant-selective inhibition through inhibitory neurons (*Fişek and Wilson, 2014*). Although several studies agree that odors are compartmentalized in the LH based on either their chemical identity (*Frechter et al., 2019*), behavioral significance (*Bates et al., 2020*; *Jeanne et al., 2018*; *Jefferis et al., 2007*), or hedonic valence (*Strutz et al., 2014*; *Tanaka et al., 2004*), it still remains controversial how the odor information is transformed from the PN to the LHN level and which odor features are coded by subtypes of LHNs.

In this study, we aimed to elucidate the odor coding and processing strategies of LHNs by investigating how a neuronal subset of particular neurotransmitter identity encodes different odor features in the LH and how this representation is correlated to their presynaptic partner neurons in the AL, the uniglomerular PNs (uPNs) and multiglomerular PNs (mPNs). Using photoactivatable GFP, we first identified diverse clusters of LHNs based on their different neurotransmitter identities and further focused our detailed analysis exclusively on glutamatergic LHNs. Using in vivo two-photon functional imaging, we characterized several aspects of odor-evoked activity in these neurons, such as odor-specific response patterns, reproducibility of repeated stimulations, as well as stereotypy across different individuals. We could successfully demonstrate that attractive and aversive odors are clearly segregated and that the response amplitudes of glutamatergic LHNs are positively correlated with the innate behavioral preference to an odor. We also dissected how the excitatory input from uPNs and odor-specific inhibition from mPNs contribute to the fine-tuning of odor-specific response patterns of LHNs. Altogether, this study demonstrates a significant role of glutamatergic LHNs regarding olfactory processing and extends our knowledge about the transformation processes of neuronal information taking place from the periphery to higher brain levels, such as the LH.

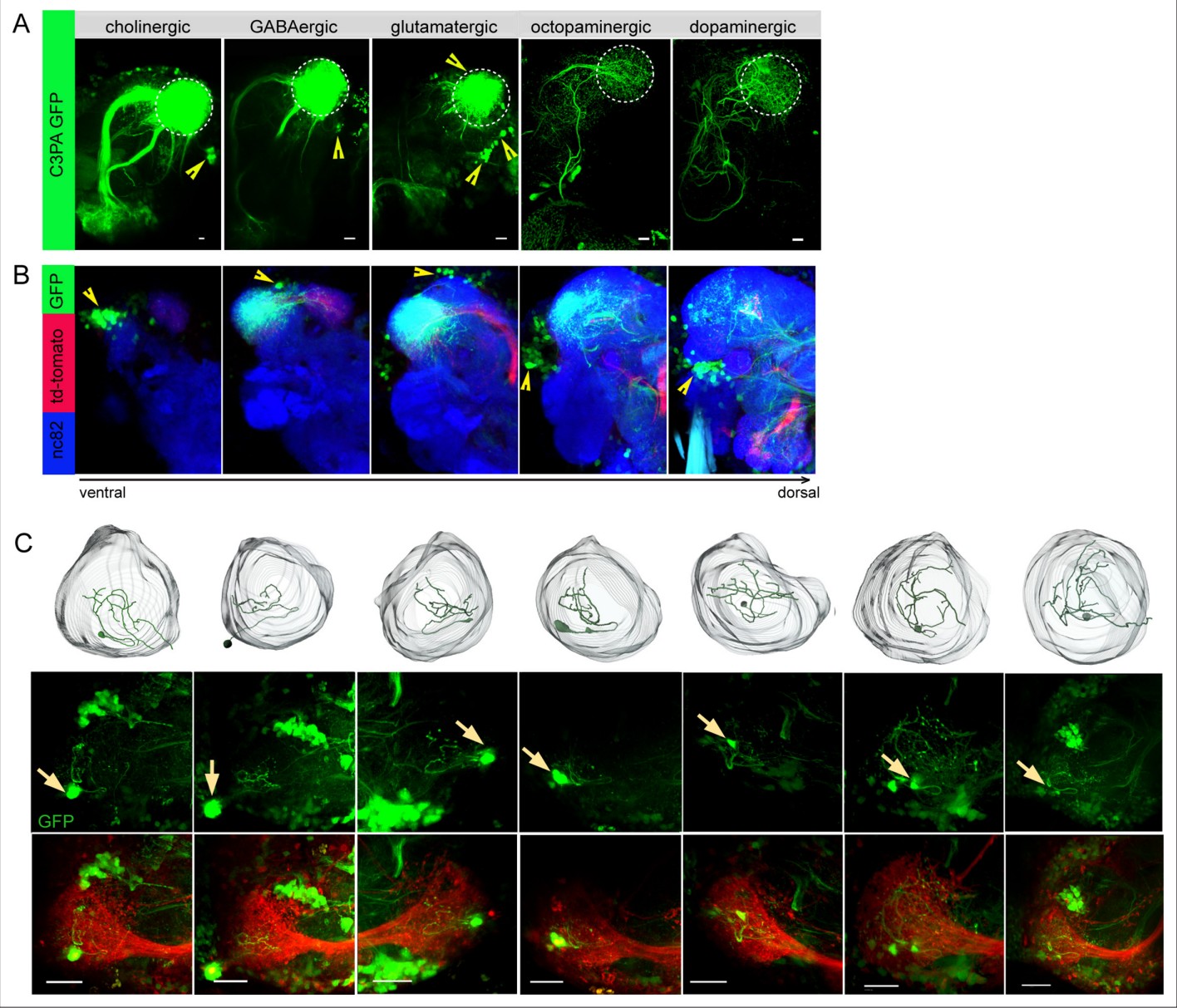

**Figure 1.** Selective labeling of lateral horn neurons (LHNs) based on their neurotransmitter identities. (**A**) Photoactivation of *UAS-C3PA-GFP* expressed under the control of different Gal4 lines with various neurotransmitter and neuromodulator identities (*Cha-Gal4, GAD1-Gal4, dVGlut-Gal4, tdc2-Gal4,* and *TH-Gal4*) reveals subsets of cholinergic, GABAergic, glutamatergic, octopaminergic, and dopaminergic LHNs (left to right). Position of cell bodies is demarcated with yellow arrowheads. (**B**) Immunohistochemistry of photoactivated brains with *dVGlut-Gal4* driving *UAS-C3PA-GFP* reveals different cell body clusters of glutamatergic LHNs found at various locations in different focal planes. *GH146-QF* driving *QUAS-mtd tomato* in the background is depicted in red and nc82 in blue. Position of the cell bodies is shown with yellow arrowheads. (**C**) Single-neuron photoactivation of the dorsomedial cluster revealed individual glutamatergic LH local neurons (LHLNs). Upper panel shows the 3D reconstruction, middle and lower panels show the photoactivated neuron in the background of *GH146-QF, QUAS mtd Tomato*. Yellow arrow demarcates the position of the photoactivated cell body. Scale bars = 10 μm.

## Results

### Identification of higher-order neurons in the LH

To unravel distinct neuronal circuits in the LH, we expressed photoactivatable GFP (*UAS-C3PA GFP*) (*Ruta et al., 2010*) under different Gal4 driver lines with various neurotransmitter and neuromodulator identities (i.e., *Cha-Gal4, GAD1-Gal4, dVGlut-Gal4, TH-Gal4,* and *tdc2-Gal4*) (*Figure 1A and B*). To precisely define the upper and lower limits of the LH, we labeled uPNs using *GH146-QF, QUAS-mtd*

*tomato* in the background as a landmark in every experiment. Using two-photon excitation, we photoactivated *C3PA-GFP* throughout the LH neuropil with laser pulses of 760 nm in the in vivo fly brain. Photoactivation of the LH using driver lines for cholinergic (*Cha-Gal4*) and GABAergic neurons (*GAD1-Gal4*) revealed clusters of LHNs with labeled somata along with cholinergic and GABAergic PNs, respectively. The labeled cholinergic and GABAergic LHN clusters were located ventrolaterally to the LH. Selective labeling of glutamatergic neurons in the LH by photoactivation using the *dVGlut-Gal4* driver line revealed several clusters of LHN cell bodies positioned dorsomedial, ventrolateral, and ventral to the LH with major neurite tracts entering the LH at distinctive locations. Glutamatergic LHNs are comprised of a large number of labeled somata (~40) and a few ventrolateral and ventral clusters possessing long neuronal tracts (*Figure 1B*). The labeled glutamatergic LHNs most likely include both LHLNs and LHONs (*Frechter et al., 2019*). Photoactivation using PA-GFP of individual neurons in the dorsomedial cluster revealed the detailed arborization of single LHLNs within the LH (*Figure 1C*). Due to the position of the cell bodies, the primary neurite, as well as the neurotransmitter identity, we assume that these LHLNs belong to the PD3a1 and PD3a2 neuron cluster described by *Dolan et al., 2019*. Photoactivation of the LH using drivers for dopaminergic (*TH-Gal4*) and octopaminergic neurons (*tdc2-Gal4*) revealed the presence of both kinds of LHNs (i.e., LHLNs, LHONs) in the LH (*Figure 1A*). These neurons innervate the LH sparsely with a very long neuronal tract and their somata positioned far away from the LH. Since we observed a high abundance of glutamatergic neurons in the LH, which were nonoverlapping with second-order neurons (i.e., PNs) deriving from the AL, we confined our further experiments on this subset of LHNs to elucidate their functional properties regarding odor coding.

## Glutamatergic LHNs reveal reproducible, stereotypic, and odor-specific responses that emerge from the PN level

To elucidate how odors are represented in the LH, we monitored odor-evoked functional responses of glutamatergic LHNs. To do so, we expressed the genetically encoded calcium sensor *UAS-GCaMP6f* (*Chen et al., 2013*) that has a high sensitivity and low basal fluorescence under the control of *dVGlut-Gal4*. In order to accurately and consistently identify a comparable focal plane for functional imaging across different animals, we expressed *QUAS-mtd tomato* in uPNs using *GH146-QF* in the background as a landmark (*Figure 2—figure supplement 1A*). We performed in vivo two-photon functional imaging from the glutamatergic LHNs at three different focal planes to comprehensively monitor odor responses throughout the LH (i.e., upper, middle, and lower planes). We used a panel of 14 ecologically relevant odors at a concentration of $10^{-3}$, which have been shown to be behaviorally either attractive or aversive (*Knaden et al., 2012*). Our imaging data show that each odor evoked a specific activity pattern across all three focal planes in the LH (*Figure 2A*). Since the LH appears rather homogenous and lacks clear morphological landmarks, we used a grid approach for imaging data analysis (*Figure 2B*) (see Materials and methods for details) and determined the ΔF/F to analyze the neuronal activity across odors for each animal. The correlation coefficient for each odor pair for the given odor panel was analyzed based on the similarity of odor responses obtained from all 65 pixels for individual animals and was then averaged across animals. At first, we compared the reproducibility of the evoked responses of these glutamatergic LHNs to two sequential presentations of each odor. This analysis of neuronal activity for repeated odor stimulations clearly showed that the odor-evoked responses were highly reproducible both qualitatively and quantitatively across trials (*Figure 2C*), which is further depicted by a high correlation coefficient across repeated stimulations for all odors (*Figure 2D*). Hence, although the odor responses appear generally broad, each odor induced an unique stereotyped response pattern in these glutamatergic LHNs (*Figure 2D*, *Figure 2—figure supplement 1B*). Furthermore, a cluster analysis based on the linkage distance between the spatial patterns of each odor response reveals that the majority of the replicates for each odor are clustered together, confirming that individual odors are represented in an odor-specific manner by glutamatergic LHNs (*Figure 2F*). To investigate whether the observed odor response properties of glutamatergic LHNs derived from their presynaptic partner neurons, we determined the odor representations of uPNs in the LH to the same odor set. Previous studies based on the innervation pattern of PNs reported a compartmentalization in the LH either based on the hedonic valence of an odor, that is, attractive versus aversive odors (*Seki et al., 2017*), the odor category, that is, pheromones versus food odors (*Jefferis et al., 2007*) or with regard to the behavioral 'odor scene' (*Bates et al., 2020*;

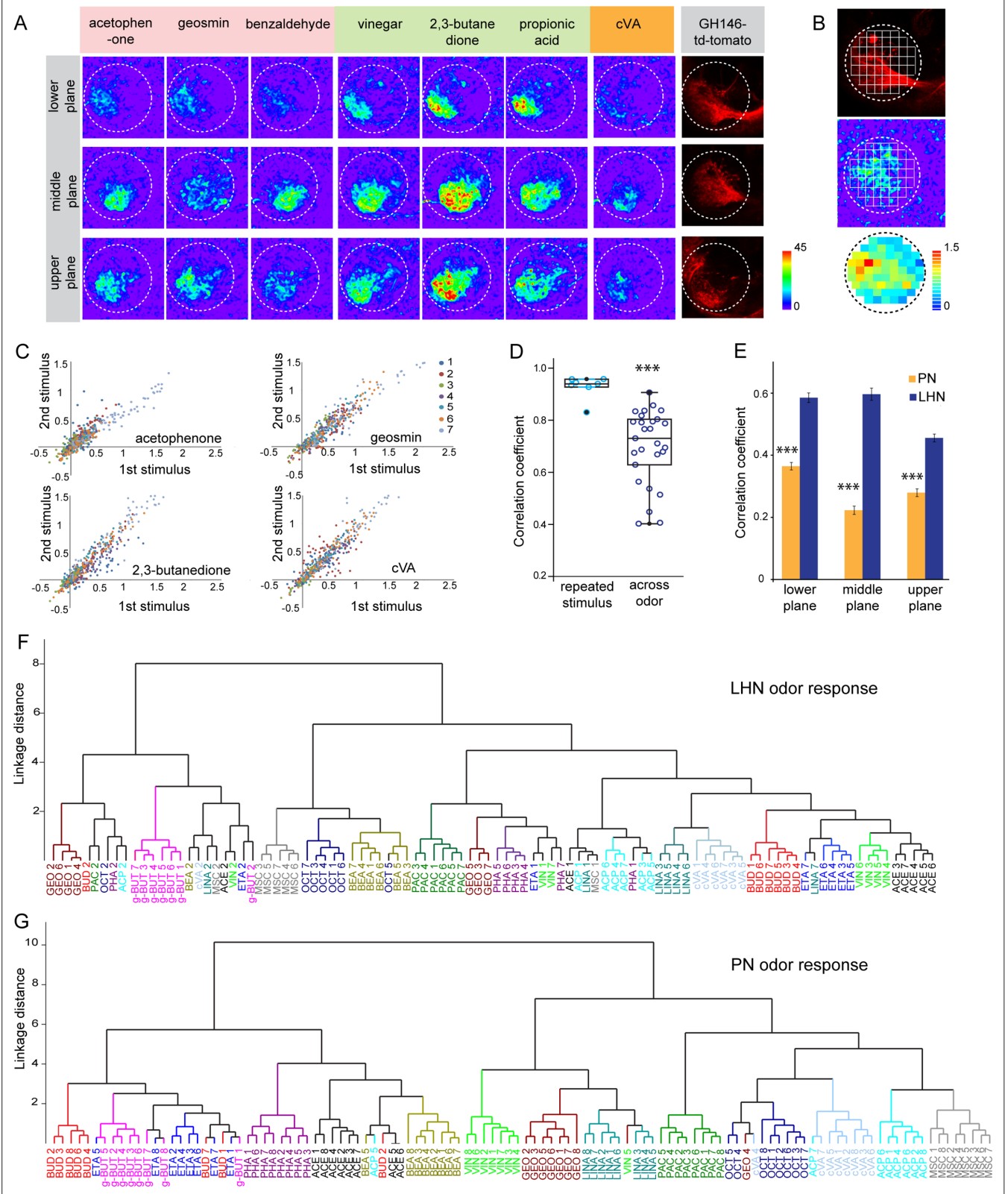

**Figure 2.** Odor response properties of glutamatergic lateral horn neurons (LHNs) and uniglomerular projection neurons (uPNs). (**A**) Representative images of calcium responses of glutamatergic LHNs in the LH brain area in three focal planes, evoked by various odors (repulsive odors: acetophenone, geosmin, benzaldehyde; attractive odors: vinegar, 2,3 butanedione, propionic acid; pheromone: cVA). Right panel shows the innervation pattern of uPNs labeled by GH146 in three different focal planes, used as a landmark to maintain comparable focal planes for functional imaging across

*Figure 2 continued on next page*

*Figure 2 continued*

different animals. (**B**) Grid approach to analyze odor-evoked responses of glutamatergic LHNs. Representative images of the grid onto uPN labeling (upper panel), used as a background landmark, odor-evoked responses of glutamatergic LHNs (middle panel) and analyzed ΔF/F of each pixels from glutamatergic LHNs in the LH area (lower panel). (**C**) Scatter plot showing calcium responses of each pixel of glutamatergic LHNs at middle plane to repeated presentation of an odor stimulus. Different colored dots depict responses obtained from different animals (n = 7). (**D**) Box plot represents comparison of odor-evoked responses of glutamatergic LHNs at middle plane, between repeated and across odor stimuli (Student's *t*-test, \*\*\*p<0.001). (**E**) Comparison of correlation coefficients of uPN and LHN responses at three different focal planes. Different odors seem to be better segregated at the uPN level than at the level of LHNs (Student's *t*-test, \*\*\*p<0.001). (**F**) Cluster analysis of LHN odor responses based on linkage distance (n = 7, ANOSIM, sequential Bonferroni significance, p=0.0001). (**G**) Cluster analysis of uPN odor responses based on linkage distance (n = 8; ANOSIM, sequential Bonferroni significance, p=0.0001). Both cluster analyses reveal that the majority of replicates for each odor are grouped together.

The online version of this article includes the following source data and figure supplement(s) for figure 2:

**Source data 1.** Summary of analyzed data to plot *Figure 2*.

**Figure supplement 1.** Stereotypy of *GH146-mtd Tomato* signal and the odor response pattern of lateral horn neurons (LHNs) and uniglomerular projection neurons (uPNs) across brains.

*Jeanne et al., 2018*). However, whether an odor-specific response map exists at the uPN level in the LH has so far not been unambiguously shown with a functional approach. We therefore expressed *UAS-GCaMP6f* under the control of *GH146-Gal4* and monitored odor-evoked responses at the two-photon microscope from the same three focal planes as used for the LHN imaging experiments. Notably, when we compared the odor response properties of uPNs for individual animals with those of LHNs we observed that the PNs possess a significantly reduced correlation coefficient as compared to LHNs, implying that odors are better segregated at the uPN level in the LH than at the third-order neuron level (*Figure 2E*). In addition, a cluster analysis based on the linkage distance between the spatial patterns of each odor response at the PN level confirms that uPNs also reveal stereotyped odor-specific response patterns in the LH (*Figure 2G*, *Figure 2—figure supplement 1C*). Altogether, these observations clearly demonstrate and signify that uPNs as well as glutamatergic LHNs exhibit an odor-specific response map in the LH.

## Response amplitude of glutamatergic LHNs reflects odor valence

Next, we investigated whether hedonic valence is still preserved at the LH level in this particular subset of LHNs as it has been described for both types of PNs (i.e., uPNs and mPNs) (*Knaden et al., 2012*; *Strutz et al., 2014*). We therefore quantified and pooled the correlation coefficients of all odor responses of glutamatergic LHNs to all attractive and aversive odors and compared those that shared the same valence (i.e., within valence), as well as for odors having an opposing valence (i.e., across valence) for all three focal planes measured (*Figure 3A*). Interestingly, we observed that the responses to odors with the same valence were significantly more similar, depicted by a higher correlation coefficient, compared to the odor responses across valence. In order to visualize the odor representations in a multidimensional space, we employed a principal component analysis (PCA) of the odor response profiles by taking into account two parameters, the spatial response patterns, as well as the intensity of odor-evoked responses by each odor. As expected, the attractive and aversive odor responses were segregated significantly throughout the LH (\*p<0.05, one-way ANOSIM), confirming that odors are indeed categorized according to their behavioral value by these third-order glutamatergic LHNs (*Figure 3B*). Notably, when only the response amplitude was considered, it was obvious that attractive odors in general evoked a stronger activity in glutamatergic LHNs compared to responses to behaviorally aversive odors across all three focal planes (*Figure 3C and D*). Next, we examined whether odor responses of second-order uPNs also reflect the hedonic valence of an odor within the LH as it has been shown for the AL level previously (*Knaden et al., 2012*). A PCA of the odor response profile of uPNs revealed that the attractive and aversive odors are clearly segregated (\*\*p<0.005, one-way ANOSIM) at the PN level also in the LH (*Figure 3E*). In order to investigate whether the response magnitude in uPNs contributes to the observed clear segregation, as observed for LHNs, we averaged the overall response activities to attractive and aversive odors across all three planes. Indeed, we observed a similar trend, meaning that uPNs responded stronger to attractive odors than to aversive ones in the LH (*Figure 3D and F*). This observation was novel and unexpected and tempted us to postulate that the flies' innate odor preference might be determined by the response amplitude of uPNs and glutamatergic LHNs. To address whether the odor-evoked response strength is correlated

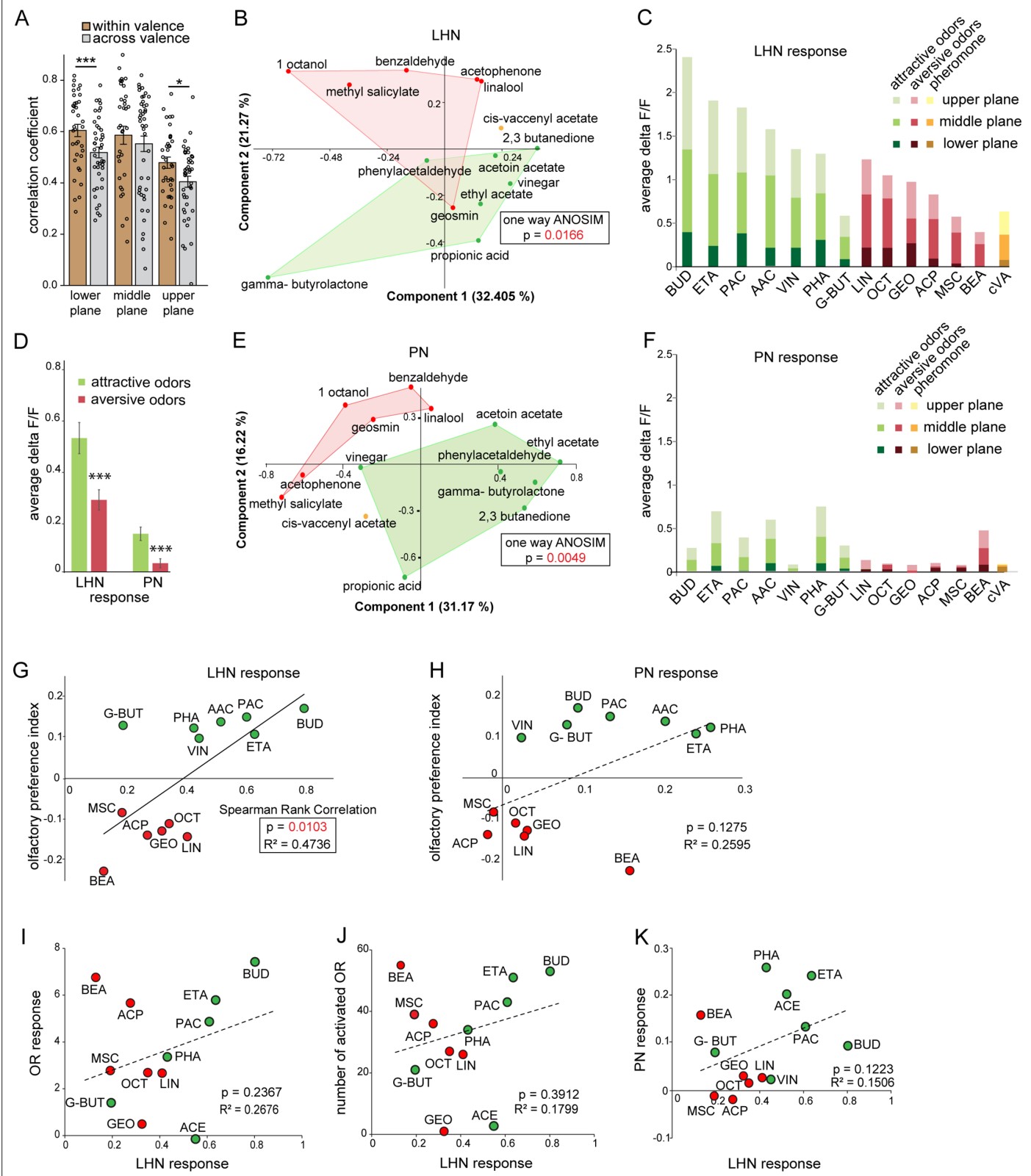

**Figure 3.** Segregation of hedonic valence in the lateral horn (LH). (**A**) Comparison of the correlation coefficients of odor responses of glutamatergic lateral horn neurons (LHNs) between the odors that share the same valence (i.e., within valence) and odors having an opposite valence (i.e., across valence). Responses to odors sharing the same valence are significantly more similar than those across valence (Student's *t*-test, ***p<0.001, *p<0.05). (**B**) Principal component analysis (PCA) of odor responses of glutamatergic LHNs shows that attractive (green) and aversive (red) odors are significantly

*Figure 3 continued*

segregated in the LH (one-way ANOSIM, p=0.0166). (**C**) Response amplitudes of glutamatergic LHNs to individual attractive and aversive odors. (**D**) Comparison of the overall response strength of glutamatergic LHNs and uniglomerular projection neurons (uPNs) to all attractive and aversive odors (Student's *t*-test, \*\*\*p<0.001). (**E**) PCA of odor responses of uPNs shows that attractive (green) and aversive (red) odors are significantly segregated in the LH (one-way ANOSIM, p=0.0049). (**F**) Response amplitudes of uPNs to individual attractive and aversive odors. (**G**) Scatter plot of olfactory preference indices of all tested odors to the response strengths of glutamatergic LHNs. The solid trend line demarcates significant correlation between the two parameters (Spearman rank correlation, p=0.01). (**H**) Scatter plot of olfactory preference indices of all tested odors to the response strengths of uPNs, which does not show a significant correlation (dotted trend line). (**I**) Scatter plot of the response strength of glutamatergic LHNs for individual odors to the overall olfactory sensory neuron (OSN) activity. (**J**) Scatter plot of the response strength of glutamatergic LHNs for individual odors to the number of ORs activated. (**K**) Scatter plot of the response strength of glutamatergic LHNs and the response strengths of uPNs for individual odors (for LHN recording n = 7 and for PN recording n = 8).

The online version of this article includes the following source data for figure 3:

**Source data 1.** Summary of analyzed data to plot *Figure 3*.

with behavioral preference, we performed T-maze assays to the same odor set as used for the functional imaging experiments. As shown in previous studies, flies displayed different levels of attraction and aversion to individual odors (*Knaden et al., 2012*; *MacWilliam et al., 2018*; *Min et al., 2013*; *Strutz et al., 2014*). Plotting the olfactory preference indices of all tested odors with the response strengths of glutamatergic LHNs indeed revealed a significant correlation (*Figure 3G*), showing that odors eliciting stronger LHN activity induce stronger attraction, while odors that elicit lower LHN responses induce a stronger aversion. Interestingly, the response strength of uPNs did not correlate significantly with the olfactory preference indices (*Figure 3H*), although uPNs activity displayed a valence-specific representation (*Figure 3C*). This observation suggests that the activity in the subset of glutamatergic LHNs may contribute significantly to the observed behavioral odor preferences. However, this correlation does not confirm any causality between the response strength of glutamatergic LHNs and the behavioral output. To test a causal relationship, it would be necessary to silence or artificially activate the glutamatergic LHNs and determine the functional consequences with regard to behavioral valence. However, since the line *dVGlut-Gal4* is broadly expressed throughout the whole brain, genetic manipulations would not be confined to the subset of glutamatergic LHNs and would therefore be difficult to evaluate. One might argue that the observed differential LHN responses emerge already at the sensory neuron level and are not the result of neuronal processing within the AL or LH. To examine this idea, we obtained odor-evoked response of olfactory sensory neurons (OSNs) to individual odors from the DoOR database (*Münch and Galizia, 2016*) and plotted the response strengths of glutamatergic LHNs either against the overall OSN activity (*Figure 3I*) or the number of ORs (olfactory receptors) activated by a specific odor (*Figure 3J*). Interestingly, we could not observe any correlation in either case, indicating that the differential LHN activity observed here is not deriving from the OSN level, but is the result of second-order neuronal processing. Notably, the comparison between uPNs and LHNs responses also did not show any correlation (*Figure 3K*), further suggesting that the differential activity observed in the glutamatergic LHNs is not a mere relay of the AL input through uPNs, but rather processed by either mPNs or other LHNs to generate the observed graded odor responses in higher-order glutamatergic LHNs.

## Polarity and connectivity of glutamatergic LHNs with uPNs and mPNs

To determine the neuronal polarity of the glutamatergic LHNs, we expressed pre- and postsynaptic markers (*UAS-sypGC3*, *UAS-homerGC3*) under the control of *dVGlut-Gal4*. Immunohistochemistry followed by confocal scanning demonstrated that these neurons possess both presynaptic dendrites and postsynaptic axon terminals in the LH (*Figure 4A*, left and right panels), indicating that the cell cluster of glutamatergic LHNs probably comprises both LHLNs and LHONs (*Dolan et al., 2019*). The presence of presynaptic terminals suggests that these neurons provide input to other LHONs, LHLNs, or even perform a feedback excitation and/or inhibition onto PNs. To assess the neuronal connectivity of these neurons with excitatory and inhibitory PNs at the morphological level in detail, we employed the GRASP technique (*Feinberg et al., 2008*). Initially, we employed two split-GFP fragments, tagged to the extracellular domain of the CD4 transmembrane protein (hereafter referred to as *CD4-GRASP*). To determine the connectivity between uPNs and glutamatergic LHNs, we expressed one fragment of GFP in glutamatergic LHNs, using *dVGlut-Gal4,* and the other fragment in uPNs, using *GH146-LexA*

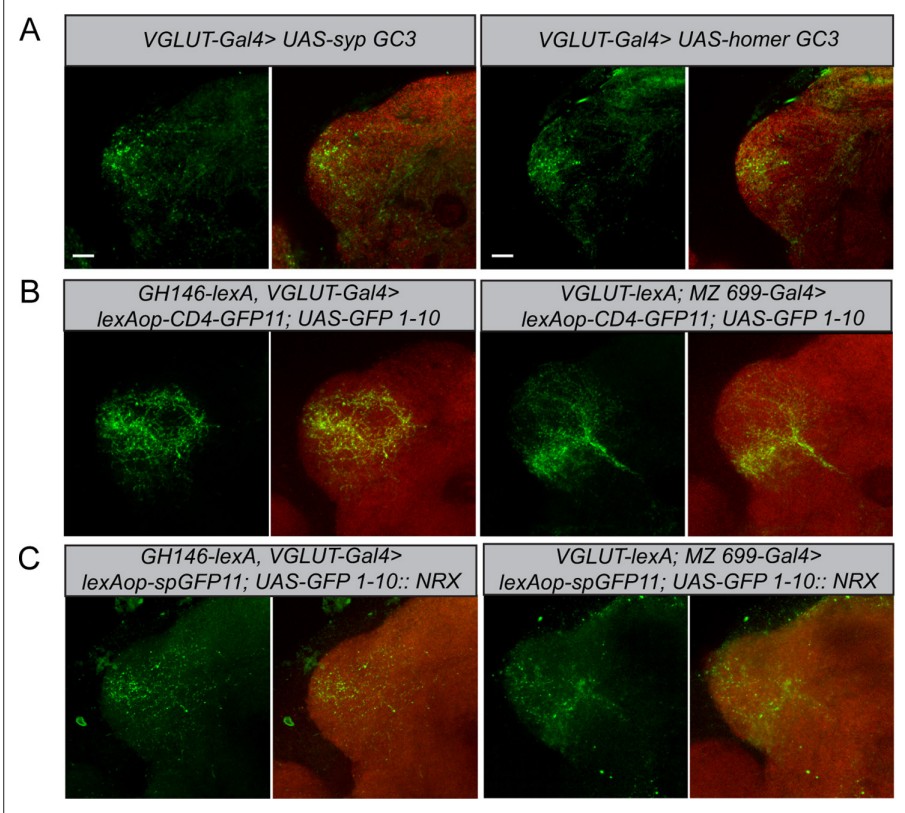

**Figure 4.** Polarity and morphological connectivity of glutamatergic lateral horn neurons (LHNs) with uniglomerular projection neurons (uPNs) and multiglomerular projection neurons (mPNs). (**A**) Expression of syp GC3 (left panel) and homer GC3 (right panel) in glutamatergic LHNs suggesting the presence of pre- as well as postsynapses in the LH. (**B**) Membrane-targeted CD4-GRASP between glutamatergic LHNs and uPNs (left panel) or mPNs (right panel). (**C**) The synaptic protein neurexin tagged to GRASP (*Nrx-GRASP*) was employed to confirm synaptic connectivity of glutamatergic LHNs with uPNs (left panel) or mPNs (right panel) in the LH. Scale bars = 10 μm.

(*Figure 4B*, left panel). Next, to examine whether mPNs and glutamatergic LHNs are synaptic partner neurons, we employed a similar approach by using *dVGlut-LexA* and *MZ699-Gal4* (*Figure 4B*, right panel). In both cases, we observed a clear neuronal connectivity to the glutamatergic LHNs. However, since *CD4-GRASP* is not synaptically targeted and can potentially lead to false-positive signals at nonsynaptic locations, we employed in addition a GFP 1–10 fragment, which is tagged to the synaptic protein neurexin, resulting in an enhancement of synaptic specificity (*Fan et al., 2013*; *Shearin et al., 2018*). Neurexin is present predominantly at the presynaptic site (*Fan et al., 2013*; *Xing et al., 2018*), but also occurs along with postsynaptic terminals (*Taniguchi et al., 2007*). When we performed GRASP by expressing neurexin in glutamatergic LHNs and the other fragment of GFP in uPNs, we observed a clear GFP signal in the LH (*Figure 4C*, left panel) implying that glutamatergic LHNs provide presynaptic output onto uPNs and therefore convey a feedback input (either excitatory or inhibitory) to the AL. Such a feedback connection needs to be verified in future experiments. In order to determine the synaptic connectivity between glutamatergic LHNs and mPNs, we expressed neurexin in mPNs and the other fragment of GFP in glutamatergic LHNs. We observed distinct GFP puncta in the LH implying that also mPNs form presynaptic and probably postsynaptic connections with glutamatergic LHNs (*Figure 4C*, right panel).

## Glutamatergic LHNs receive major excitatory input from uPNs and an odor-specific inhibition from mPNs

In order to further elucidate the neural connectivity and verify our GRASP experiments, we determined the functional connectivity of glutamatergic LHNs with uPNs and mPNs by transection experiments using two-photon laser-mediated microdissection. To manipulate the input from uPNs, we

microlesioned the iACT tract entering the LH and compared the odor-evoked responses to vinegar and benzaldehyde from the glutamatergic LHNs before and after laser ablation (*Figure 5*, schematic upper panel). After transecting the iACT tract, we observed that the odor-induced responses to both vinegar and benzaldehyde were completely abolished across all three focal planes of the LH (*Figure 5A–A″ and B–B″*). This finding implies that glutamatergic LHNs receive their major excitatory input from the AL through uPNs. To verify whether this complete elimination of the odor-evoked activity is not just a technical side effect of the laser ablation procedure, we also monitored odor responses from the glutamatergic LHNs in the other, untreated brain hemisphere before and after microlesioning, which proved not to be affected (*Figure 5—figure supplement 1*). In order to examine the functional connectivity between mPNs and glutamatergic LHNs, we also transected the mACT tract entering the LH and monitored odor responses from glutamatergic LHNs before and after laser transection. Notably, transecting the mACT led to a significant increase in the odor-evoked responses in the middle and upper planes of the glutamatergic LHNs (*Figure 5C–C″ and D–D″*). We further observed that the mACT transection affected the odor-evoked responses differentially in a way that the response increase was higher in the case of vinegar than for benzaldehyde. In order to investigate whether this response increase following a transection of the mACT is odor-specific, we performed this treatment with four other odors, two attractive ones (2,3 butanedione, ethyl acetate) and two aversive odors (acetophenone, linalool). Notably, only acetophone revealed a significant increment in the odor-evoked responses of the glutamatergic LHNs after transection, while we observed only a trend for 2,3 butanedione, ethyl acetate, and linalool, which was not significantly different from the odor responses before transection (*Figure 5—figure supplement 2*). These observations altogether indicate that the glutamatergic LHNs receive an odor-selective inhibition from mPNs which is stronger for some odors, such as vinegar and acetophenone, moderate for benzaldehyde, and very weak or absent for others.

## mPN-mediated inhibition onto glutamatergic LHNs contributes to odor specificity

Next, we aimed to understand the impact of the observed odor-specific inhibition of mPNs with regard to the odor-specific responses of glutamatergic LHNs in the LH. To address this issue, we monitored odor-evoked responses in the glutamatergic LHNs to two repeated stimulations of vinegar and benzaldehyde (depicted as 1 and 2 in the schematic in *Figure 6A and A′*), followed by measuring LHN activity after microlesioning the mACT tract (depicted as 3 in *Figure 6A and A′*). Since, we observed a significant response increase following mPN-mediated inhibition only in the middle and upper planes (*Figure 5C′, C″, D′ and D″*), we confined our analysis to these two areas. In order to determine whether the laser ablation of mACT alters also the odor-evoked spatial pattern in addition to an increase in the response amplitude, we compared the correlation coefficients between the repeated responses before laser ablation to those after the transection. Our results revealed that removal of the inhibitory input from mPNs leads to a reduced correlation coefficient of the odor response patterns evoked by vinegar in LHNs (*Figure 6B and B′*). Interestingly, such an effect was not observed in the case of benzaldehyde, although we observed a weak mPN-mediated inhibition for this odor before (*Figure 6B and B′*). Since the odor-evoked spatial patterns were modified after silencing the mPN input into the LH, we next wondered whether this also affected the odor specificity of the LHN responses. We therefore compared the correlation coefficients 'across odors' (i.e., between vinegar and benzaldehyde) before and after microlesion (*Figure 6C*). Strikingly, we observed that the odor-evoked response patterns of vinegar and benzaldehyde became increasingly similar after removal of the mPN-mediated inhibition depicted as an increased correlation coefficient after laser ablation. This finding suggests that the inhibitory input from mPNs onto glutamatergic LHNs is crucial to generate a distinct and odor-specific response pattern for individual odors at the LH level.

## Discussion

Our study functionally characterizes a subset of glutamatergic higher-order neurons in the LH regarding odor coding and processing. We demonstrate that glutamatergic LHNs respond in a reproducible, stereotypic, and odor-specific manner and these response properties emerge at the level of presynaptic uPNs (*Figure 6D*). Notably, the differential activity levels of glutamatergic LHNs to attractive and

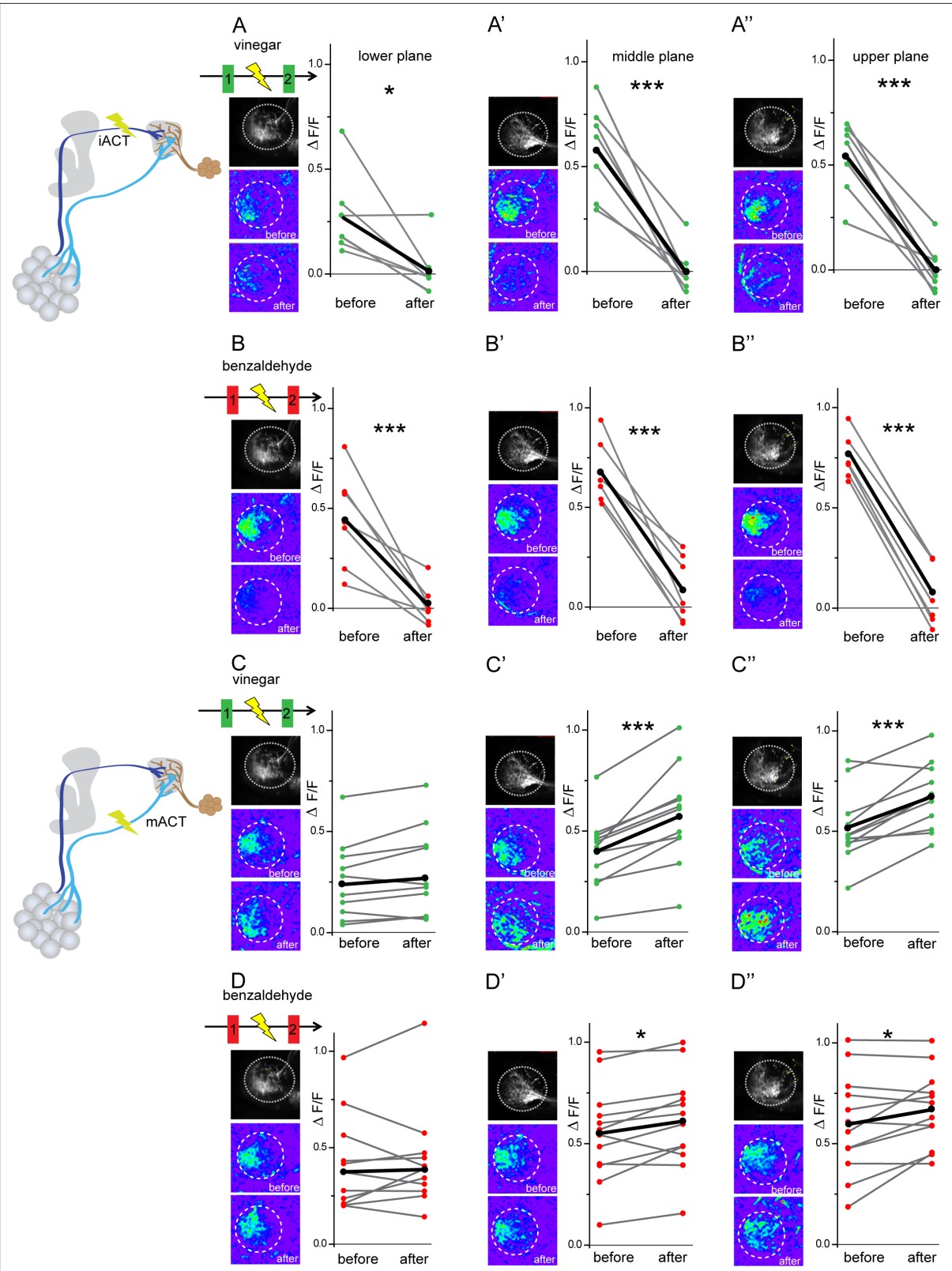

**Figure 5.** Glutamatergic lateral horn neurons (LHNs) receive their major excitatory input from uniglomerular projection neurons (uPNs) and an odorant-selective inhibition from multiglomerular projection neurons (mPNs). Upper panel schematic represents the experimental approach: the iACT tract was laser transected while odor-evoked responses were monitored from glutamatergic LHNs. (**A, A', A''**) Representative images and graphical comparison of responses evoked by vinegar in glutamatergic LHNs before and after laser transection of the iACT, across three different planes (n = 6). (**B, B',**

*Figure 5 continued on next page*

*Figure 5 continued*

**B″**) Representative images and graphical comparison of responses evoked by benzaldehyde in glutamatergic LHNs before and after laser transection of the iACT (n = 6). Lower panel schematic represents the experimental approach: the mACT tract was laser transected while odor-evoked responses were measured from glutamatergic LHNs. (**C, C′, C″**) Representative images and graphical comparison of vinegar-evoked responses of glutamatergic LHNs before and after laser transection of the mACT (n = 10). (**D, D′, D″**) Representative images and graphical comparison of benzaldehyde-evoked responses of glutamatergic LHNs before and after laser transection of the mACT (n = 10) (paired *t*-test ***p<0.001, *p<0.05). The red and green circles indicate individual data points for benzaldehyde and vinegar, respectively. The black circles indicate the averaged response.

The online version of this article includes the following source data and figure supplement(s) for figure 5:

**Source data 1.** Summary of analyzed data to plot *Figure 5*.

**Figure supplement 1.** Laser transection does not affect activity of glutamatergic lateral horn neurons (LHNs) in the intact brain hemisphere.

**Figure supplement 1—source data 1.** Summary of analyzed data to plot *Figure 5—figure supplement 1*.

**Figure supplement 2.** Multiglomerular projection neuron (mPN)-mediated odor-selective inhibition onto glutamatergic lateral horn neurons (LHNs).

aversive odors are positively correlated to the olfactory behavioral preference, indicating that these neurons are mainly tuned to attractive odors. The response features do not arise from the OSN level, but rather derive from local processing within the LH by integrating inputs from multiple olfactory channels through uPNs, which also show a valence-specific odor representation. Furthermore, laser transection experiments demonstrate that these higher-order neurons receive their major excitatory input from uPNs and an odor-specific inhibitory input from mPNs. Lastly, our data show that the observed mPN-mediated inhibition seems to be required for generating an odor-specific response map in the LH (*Figure 6D*).

## Glutamatergic LHNs respond in a stereotypic and odor-specific manner

A growing body of evidence suggests the existence of an anatomical and functional stereotypy in early processing centers of the insect olfactory pathway (*Hildebrand and Shepherd, 1997*; *Korsching, 2002*). This stereotypy becomes obvious first at the sensory neuron (OSN) level, where OSNs expressing a certain OR target and converge on a stereotypic glomerulus, resulting in a conserved spatial map in the AL between different individuals (*Wang et al., 2003*; *Wilson et al., 2004*). This anatomical stereotypy was shown to be retained at the postsynaptic PN level (*Jefferis et al., 2007*; *Tanaka et al., 2004*).

Several studies support the notion that an anatomical stereotypy might also be present at the level of the LH, particularly shown for the PN to LHN connectivity (*Bates et al., 2020*; *Fişek and Wilson, 2014*; *Jeanne et al., 2018*; *Jefferis et al., 2007*). Along this line, functional studies have demonstrated that LHNs respond in a reproducible and stereotyped manner to odors and this stereotypy is a general feature of the LH (*Fişek and Wilson, 2014*; *Frechter et al., 2019*). However, how an ensemble of LHNs integrates inputs from several olfactory channels, that is, the presynaptic excitatory and inhibitory PNs, and whether each odor induces a specific and stereotyped response pattern in the LH was not clearly addressed before. In our study, we demonstrate that each odor is represented by an odor-specific activity pattern in the LH, while the examined glutamatergic LHNs display broader tuning patterns than their presynaptic partner neurons. Although it has been assumed previously that odor specificity may not be encoded in higher-order brain centers (*Grabe and Sachse, 2018*; *Strutz et al., 2014*), our findings are in accordance with a recent study by *Frechter et al., 2019*, who demonstrate the existence of 33 different LH cell types exhibiting stereotypic odor response properties with increased tuning breadth than PNs. The observed odor-evoked response features of LHNs that are odor-specific but with a broader tuning breadth could be due to several reasons. First, they receive input from similarly as well as differently tuned uPNs; therefore, the topographic map of uPN axonal terminals is not clearly retained at the level of LHNs. Second, LHNs integrate inputs from multiple odor channels, for example, one LHN receives on average excitatory input from ~5.2–6.2 glomeruli (*Frechter et al., 2019*; *Jeanne et al., 2018*). Third, both uPNs and mPNs provide input to glutamatergic LHNs and those LHNs in turn feedback onto those second-order neurons and provide feedforward information to other LHNs (*Bates et al., 2020*; *Frechter et al., 2019*; *Jeanne et al., 2018*; *Figure 6D*). Here, we observed that uPNs are more efficient in encoding odor identity than the glutamatergic LHNs in the LH (*Figure 6D*), whereas LHNs reveal an improved categorization of odors either based on behavioral significance, 'odor scene' or chemical group (*Bates et al., 2020*; *Frechter*

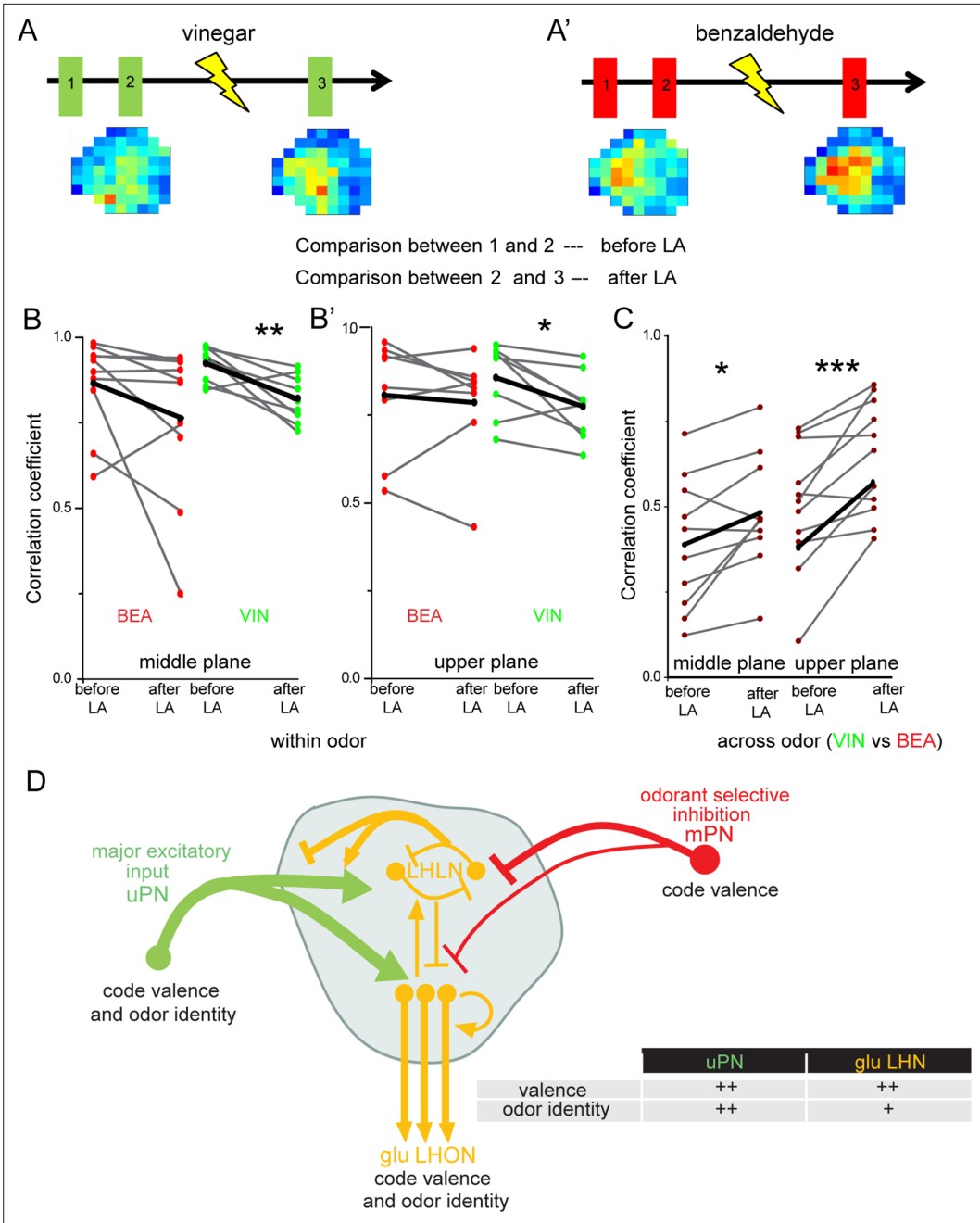

**Figure 6.** Multiglomerular projection neuron (mPN)-mediated inhibition facilitates odor specificity in glutamatergic lateral horn neurons (LHNs). (**A, A'**) Schematic representing the experimental approach: odor-evoked responses of glutamatergic LHNs were monitored to repeated odor presentations (1 and 2) before and after laser transection of the mACT (3) for the odors vinegar and benzaldehyde. (**B, B'**) Comparison of the correlation coefficients between repeated odor responses before laser ablation to those after the transection for vinegar and benzaldehyde (within odor comparison) (n = 7–9). (**C**) Comparison of the correlation coefficients across vinegar and benzaldehyde before and after microlesion (across odor comparison; paired *t*-test, ***p<0.001, **p<0.005, *p<0.05) (n = 10–11). (**D**) Schematic summarizing the observed connectivity between PNs and glutamatergic LHNs in the LH: glutamatergic LHNs consist of both LH local neurons (LHLNs) and LH output neurons (LHONs). Uniglomerular projection neurons (uPNs) (green) provide the major excitatory input to glutamatergic LHNs (yellow) while they also receive feedback input (either excitation or inhibition) from LHLNs (**Bates et al., 2020**). LHLNs have been reported to inhibit each other as well as LHONs as well (**Bates et al., 2020**). LHONs, in addition to providing feedback excitation to LHLNs, relay the information to further higher brain centers. mPNs provide an odorant-selective inhibition to glutamatergic LHNs. mPNs are known to encode odor valence in the LH (**Strutz et al., 2014**),

*Figure 6 continued on next page*

*Figure 6 continued*

uPNs and glutamatergic LHNs encode odor identity as well as odor valence. The thickness of the line indicates the strength of excitatory or inhibitory input. The table illustrates that uPNs encode an improved odor identity than postsynaptic glutamatergic LHNs, whereas hedonic valence is equally maintained at the level of uPNs and glutamatergic LHNs in the LH.

The online version of this article includes the following source data for figure 6:

**Source data 1.** Summary of analyzed data to plot *Figure 6*.

*et al., 2019*; *Jeanne et al., 2018*). However, the distinct odor-specific response map by glutamatergic LHNs observed in our study suggests that the dimensionality of odor features might not get reduced but still retains information about the odor specificity at the third-order processing stage.

## Coding of hedonic odor valence

Our functional imaging recordings revealed that odor valence is encoded by glutamatergic LHNs, leading to different activation strengths and patterns for attractive and aversive odors in the LH. Although the valence code is already present at the AL level in uPNs (*Knaden et al., 2012*), it could be assumed that the same valence code might be translated to the next level of higher-order neurons. However, evidence of high convergence and divergence of neurons from different sensory modalities in the LH argues against a simple translation of the valence code or odor identity to the LHN level (*Huoviala et al., 2018*). Our observation is well in line with previous studies that have revealed that odor-evoked responses in higher brain centers are generally categorized according to certain odor features as already mentioned above. For example, using functional imaging or patch-clamp recordings of second-order olfactory neurons revealed the existence of distinct attractive and aversive odor response domains in the LH formed by uPNs and mPNs (*Seki et al., 2017*; *Strutz et al., 2014*). Such a categorization according to hedonic valence is also visible in our study when the odor-evoked responses of glutamatergic LHNs were plotted in a PCA, taking into account the spatial response patterns as well as the intensity of activity. Although no prominent spatial domain of attractive or aversive odors was evident in our recordings, we observed that attractive odors evoked a generally stronger activity when compared to aversive odors in this subset of LHNs. We noted a similar trend in second-order uPNs. However, their response strength was neither correlated with the olfactory preference determined in behavior nor the odor response properties of LHNs. The observed significant correlation between the amount of odor-evoked activity in glutamatergic LHNs to the behavioral valence of an odor leads us to postulate that the activity strengths of higher-order olfactory neurons to odor stimulation might determine the behavioral response – an assumption that needs to be tested in future studies.

Although we have documented in our study how the glutamatergic LHNs determine the innate behavioral valence, odor valence in the LH can also be achieved through learning and LHNs have also been shown to play a critical role for learned behavior. A specific class of LHNs (so-called PD2a1/b1) has been reported to mediate innate approach response in addition to learned avoidance response in an odor concentration-dependent manner (*Das Chakraborty and Sachse, 2021*; *Dolan et al., 2018*). Therefore, depending upon the context, the same LHNs can mediate innate as well as learned behavior with opposing valence. Extensive interconnections between the two higher brain centers, the MB and the LH through MBONs and LHONs, signify how the MB modulates the innate olfactory pathways and the valence code in the LH (*Bates et al., 2020*; *Das Chakraborty and Sachse, 2021*; *Dolan et al., 2019*; *Frechter et al., 2019*).

Our study suggests that glutamatergic LHNs use different strategies to extract different features of odor information, (1) conserving the identity of an olfactory stimulus by forming an odor-specific activity map and (2) encoding the valence of an odor by integrating information from multiple olfactory channels. It is an ongoing debate regarding how neurons in the LH evaluate an odor stimulus. Gradual detection, encoding, and categorization of an odor at different olfactory processing levels can result in a simple binary choice or complex behavioral responses of an animal. The behavioral preferences are simply reflected in either to approach (positive) or to leave (negative), to copulate (positive) or to reject (negative), and to oviposit (positive) or to find another suitable oviposition site (negative) depending upon the behavioral assays (*Galizia, 2014*). Hence, based on the context or ecological relevance, an odor can be evaluated either as 'pleasant' or 'unpleasant,' which is well

reflected by the response properties of glutamatergic LHNs regarding their valence specificity and their correlation between response strength and behavioral odor preference.

Notably, such a correlation between response intensity and behavioral preference has also been observed in previous studies, where the amplitude of food odor-evoked activity in neuropeptide F (dNPF) neurons was found to strongly correlate with food odor attractiveness (*Beshel and Zhong, 2013*). Another study that combined functional imaging with tracking of innate behavioral responses revealed that the behavioral output could be accurately predicted by a model summing up the normalized glomerular responses, in which each glomerulus contributes a small but specific part to the resulting odor preference (*Badel et al., 2016*). At the level of the LH, LHNs then integrate the olfactory information from the glomerular responses conveyed via uPNs and mPNs. In one of our previous studies, we demonstrated that mPNs respond differently to attractive and aversive odors, and mediate behavioral attraction (*Strutz et al., 2014*). In this context, our study complements this previous finding by showing that also uPNs display distinct valence-specific responses in the LH to attractive and aversive odors. Information from these two PN pathways becomes integrated and processed in the LH, resulting in valence-specific activities in glutamatergic LHNs, which may in turn determine the relative behavioral preference.

## mPN-mediated inhibition generates odor-specific response patterns in the LH

We demonstrate that mPNs inhibit the glutamatergic LHNs in an odor-selective manner, leading to an odor-specific response pattern. According to our observations, glutamatergic LHNs receive a stronger inhibition from mPNs in response to the odors vinegar and acetophenone than to benzalde-hyde, whereas other odors, such as 2,3 butanedione, linalool, and ethyl acetate, seem not to induce an inhibition. In the absence of this inhibition, we noted that in addition to an increased response amplitude and altered odor representation, the activity patterns of different odors became more strongly correlated and hence more similar. We therefore conclude that the mPN-mediated selective inhibition on this glutamatergic subset of LHNs is necessary to maintain odor specificity. Along this line, a previous study has reported that mPNs provide an odor-selective input to vlpr neurons, another class of third-order LHNs (*Liang et al., 2013*). According to these authors, this odor-specific modulation depends on the nature of the odor and results from the stereotyped connectivity of mPNs in the AL as well as in the LH. Although this study provides evidence that uPNs are not presyn-aptically inhibited by mPNs, another study established that mPNs indeed inhibit uPNs in the LH, facilitating odor discrimination (*Parnas et al., 2013*). In addition to uPNs and vlpr neurons, our study identifies another class of recipient neurons (glutamatergic LHNs) that receives mPN-mediated odorant-selective inhibition.

The glutamatergic LHN population in our study comprises glutamatergic LHONs as well as LHLNs since we used a broad line that labels an ensemble of all glutamatergic LHNs. The study by *Dolan et al., 2019* employed specific split-Gal4 lines to selectively label LHINs, LHLNs, and LHONs with different neurotransmitter identities and analyzed their connectivity using EM connectomics. By employing artificial activation of specific subsets of LHNs via CsChrimson, *Dolan et al., 2019* was able to demonstrate that one class of glutamatergic LHLNs (so-called PV4a1:5) forms excitatory synapses with AV1a1 LHONs that mediate aversive behavioral responses. In our study, since we used a generic Gal4-line to label all glutamatergic LHNs, we could neither activate nor silence specific neuronal subsets to observe their relevance with regard to odor-guided behavior. However, when we correlated the odor response strength of LHNs to the olfactory preference, we observed that these neurons are mainly tuned to attractive odors, suggesting that they are involved in mediating approach behavior. Although we were not able to clearly differentiate the functional properties between different popu-lations of LHONs and LHLNs, our study provides the first understanding of how odors are integrated, transformed, and finally represented in the LH by an ensemble of glutamatergic LHNs.

Intriguingly, the neurotransmitter identity of this class of LHNs opens up another interesting aspect: Knowing that glutamate can act as an excitatory (*Aungst et al., 2003*; *Das et al., 2011a*) or inhibi-tory neurotransmitter (*Liu and Wilson, 2013*), as well as a coincident detector (*Das et al., 2011b*), depending upon the receptors present in the postsynaptic neurons, further experiments are needed to reveal the consequences of glutamatergic LHN input onto their postsynaptic partner neurons. Certainly, the presence of an impressive amount of vesicular glutamate in the LH points towards a

significant role of glutamatergic LHNs with regard to odor coding and processing at this higher brain center.

## Materials and methods

### Fly stocks

Flies were raised on autoclaved cornmeal-yeast-sucrose-agar food in a 12 hr light/dark cycle at 25°C incubator. The following lines have been used for functional imaging: *dVGlut-Gal4* (II) (*Mahr and Aberle, 2006*) (Bloomington 26160), *GH146-Gal4* (II) (*Stocker et al., 1997*) (from Leslie Vosshall's lab), *GH146-QF, QUAS mtd Tomato* (BDSC 30037), and *UAS-GCaMP6f* (III) (*Chen et al., 2013*) (Bloomington 52869). For photoactivation experiments, *Cha-Gal4* (II) (Bloomington 6798), *GAD1-Gal4* (II) (*Ng, 2002*) (Bloomington 51630), *dVGlut-Gal4*, *TH-Gal4* (III) (*Zhang et al., 2007*) (from Mani Ramaswami's lab), *tdc2-Gal4* (II) (*Koon et al., 2011*) (NCBS, India), *UAS-syp GCaMP3* (*Pech et al., 2015*), *UAS-homer GCaMP3* (*Pech et al., 2015*) (gift from André Fiala), and *UAS-C3PA* (*Ruta et al., 2010*) (gift from Sandeep Datta) flies have been used. Wildtype *Canton-S* flies have been used for behavioral experiments.

### Photoactivation

*UAS-C3PA* was expressed under *Cha-Gal4, GAD1-Gal4, dVGlut-GAL4, TH-Gal4,* and *tdc2-Gal4* in the background of *GH146-QF, QUAS mtd Tomato* for the photoactivation experiment. An initial pre-photoactivation scan of the whole LH area was taken at 925 nm with a ×40 water immersion objective. The extent of the LH in z-section was identified based on the innervation pattern of uPNs (visualized by *GH146-QF, QUAS mtd Tomato*). A region of interest in different focal planes covering the entire LH area was photoactivated for ~1–2 min followed by 2 min rest using ~7 mW of 760 nm of laser. We allowed 10–15 min for photoactivated GFP to diffuse in more distal neural processes. The post-photoactivation scan was taken using the same set-up as used for the pre-photoactivation scan.

For the single-neuron photoactivation experiment, we used a small region of interest to cover only a single-cell body in the dorsomedial cluster of glutamatergic LHNs. We could not photoactivate the cell bodies from other clusters since they were difficult to locate without photoactivation of the entire LH. After brief laser pulses using ~7 mW of 760 nm for 5–8 s, the GFP labeling in the photoactivated neuron was monitored. After repeating brief photoactivation, the final post-photoactivation scan was obtained.

### AMIRA 3D reconstruction

We performed the single-neuron 3D reconstruction using AMIRA 5.6 software. The red channel with *GH146-QF, QUAS mtd Tomato* was used to mark the LH. The workflow to label the LH was as follows: labeling > label field > marking the boundary with paint brush and extrapolate (ctrl+I) to cover the entire lateral horn > add to the object > surfacegen > unconstrained smoothening > surfaceview > transparent. To reconstruct the photoactivated single neuron, the following workflow was used. Orthoslice of the GFP channel to visualize the photoactivated neuron > Create > skeleton > skeleton tree > graph editor > select a point of the photoactivated cell body > add a new point or branch to label accurately the entire photoactivated neuron. Different parameters such as diameter, lines, surfaces, and colors can be modified in skeletonview to obtain the best visual representation.

### Functional imaging

All functional imaging experiments were performed on 4–7-day-old mated female flies. Flies were dissected for optical imaging according to the protocol by *Strutz et al., 2014*. Flies were briefly immobilized on ice and then mounted onto a custom-made Plexiglas stage with a copper plate (Athene Grids, Plano). A needle before the head was placed to stabilize the proboscis and align the head properly. Protemp II composite (3M ESPE) was used to fix the head with the copper plate. We bent the anterior part of the fly's head with fine gold wire, and a small plastic plate having a round window was placed on top. We sealed the head with that plate using two-component silicone (Kwik Sil) and leaving the center part open to make a cut. The cuticle between the eyes and the ocelli was gently cut under saline (130 mM NaCl, 5 mM KCl, 2 mM MgCl$_2$, 2 mM CaCl$_2$, 36 mM saccharose, 5 mM

HEPES, 1 M NaOH, pH 7.3). The cuticle was either bent forward and fixed to the silicon or removed. All fat, trachea, and air sacs were removed carefully.

Functional imaging was performed using a two-photon laser scanning microscope (2PCLSM, Zeiss LSM 710 meta NLO) equipped with an infrared Chameleon Ultra diode-pumped laser (Coherent, Santa Clara, CA) and a ×40 water immersion objective lens (W Plan-Apochromat ×40/1.0 DIC M27). The microscope and the laser were placed on a smart table UT2 (New Corporation, Irvine, CA). The fluorophore of GCaMP6f was excited with 925 nm. For each individual measurement, a series of 40 frames (corresponding to 10 s) acquired at a resolution of 256 × 256 pixels was taken with a frequency of 4 Hz. During the entire 10 s of recording, each odor was delivered after 2 s (eight frames) for 2 s (eight frames). To cover the entire LH, images were acquired from three different focal planes (i.e., upper, middle, and lower), each 15 µm apart in z-direction. *GH146-QF, QUAS-mtd tomato* was always used in the background to precisely locate the planes for all imaging experiments. A set of 14 odors was used, including phenylacetaldehyde (PHA), vinegar (VIN), 2,3 butanedione (BUD), acetoin acetate (AAC), ethyl acetate (ETA), propionic acid (PAC), and gamma-butyrolactone (G-BUT) as attractive odors, 1-octanol (OCT), benzaldehyde (BEA), acetophenone (ACP), methyl salicylate (MSC), geosmin (GEO), and linalool (LIN) as aversive odors, and *cis*-vaccenyl acetate (cVA) as a pheromone. All odors have been used at a concentration of $10^{-3}$ diluted in mineral oil. Vinegar was diluted in double-distilled water. Flies were imaged for up to 1 hr, with a minimum interstimulus interval of 1 min. For odor delivery, we used a computer-controlled odor delivery system (described in *Mohamed et al., 2019*).

## Data and statistical analysis for functional imaging

The LSM file obtained from the imaging software ZEN was processed and analyzed in Fiji. 40 frames for each odor (40 frames × 14 odors) were opened and stitched together using 'Image > Stacks > Tools > Concatenate' command. The file combining all the frames for all odors was movement-corrected using 'Plug in > Registration > Stackreg > Rigid body' command. The entire LH area was encircled and cropped and divided into 65 pixels, each having 30 × 30 µm² dimension. Using 'Analyze > Tools > ROI manager > more > multi measure' command, fluorescence intensity for every pixel for every frame for every odor was obtained. This huge dataset was further processed using Excel software. Average ΔF/F was calculated for 10–18 frames for each pixel for each odor. The ΔF/F for 65 pixels for 14 odors for each replicate was aligned in table format and processed using 'past' statistical software. The response amplitude of all 65 pixels of individual brains was compared between each odor pair for the entire odor panel to obtain the correlation coefficient using Euclidean distance. The response amplitude of each grid was compared only within one brain and not across brains. The obtained correlation coefficients from individual animals were then averaged and compared for odor reproducibility, odor specificity or between LHNs and PNs.

## Two-photon-mediated laser transection

Transection of either the iACT or mACT tract of PNs was performed using *GH146-QF, QUAS-mtd tomato* as a guiding landmark. Transection of the PN tracts was done in one brain hemisphere of each fly only. The chosen area for transection was located using 925 nm of laser, close to the LH but far enough not to damage the neurites branching in the LH neuropil. Approximately 60 mW of 760 nm laser was provided at the chosen area at a pulse for 2–3 s for 2–3 times with 1 s time interval (*Strutz et al., 2014*). Successful transection was verified after visualization of a small bubble. As a control, the LH from the other, untreated brain hemisphere was imaged to stimulation with benzaldehyde and 2,3 butanedione before and after laser ablation to rule out that the laser transection of the specific targeted area caused any nonspecific damage to other brain areas.

## Behavioral assay

T-maze experiments with 4–7-day-old flies were carried out for the olfactory preference assay. Approximately 30 flies were starved for 24 hr in each vial containing wet filter paper. Mated males and females were used for the behavioral assay. All odors were used at a concentration of $10^{-3}$, which is the same as used for the functional imaging experiments. At each arm of the T-maze, small filter papers containing either 6 µl of mineral oil (control) or an odor (odorant) were placed. Briefly cold anesthetized flies were released at the junction of horizontally placed T-maze. After ~30 min, flies in each side of the arm were

counted. The preference index was calculated as (odorant-control)/total number of flies (i.e., 30). Each odor has ~20 replicates.

## Acknowledgements

This research was supported through funding by the Max Planck Society and Alexander von Humboldt Foundation (grant to SDC). Stocks were obtained from the Bloomington Drosophila Stock Center and used in this study (NIH P40OD018537). We express our gratitude to Silke Trautheim for her excellent support in fly rearing. We thank to Veit for his expert help regarding image analysis and Sonja for her valuable suggestions regarding statistical analysis.

## Additional information

### Funding

| Funder | Grant reference number | Author |
|---|---|---|
| Max Planck Society | | Sudeshna Das Chakraborty Hetan Chang Bill S Hansson Silke Sachse |
| Alexander von Humboldt-Stiftung | | Sudeshna Das Chakraborty |

The funders had no role in study design, data collection and interpretation, or the decision to submit the work for publication.

### Author contributions

Sudeshna Das Chakraborty, Conceptualization, Formal analysis, Funding acquisition, Investigation, Methodology, Visualization, Writing – original draft, Writing – review and editing; Hetan Chang, Formal analysis, Investigation; Bill S Hansson, Funding acquisition, Supervision, Writing – review and editing; Silke Sachse, Conceptualization, Formal analysis, Funding acquisition, Project administration, Supervision, Visualization, Writing – original draft, Writing – review and editing

### Author ORCIDs

Sudeshna Das Chakraborty ⓘ http://orcid.org/0000-0002-4700-3511
Bill S Hansson ⓘ http://orcid.org/0000-0002-4811-1223
Silke Sachse ⓘ http://orcid.org/0000-0002-9769-8067

### Decision letter and Author response

Decision letter https://doi.org/10.7554/eLife.74637.sa1
Author response https://doi.org/10.7554/eLife.74637.sa2

## Additional files

### Supplementary files

• Transparent reporting form

### Data availability

All data generated and analysed in this study are included in the manuscript and supporting file. We provide all source data of this study. The fluorescent changes of GCaMP6f obtained by 2-photon functional imaging deriving from the experiments that are represented as graphs in the main and figure supplements are provided as excel files (Figs. 2, 3, 5, 6). All raw data files of the functional imaging experiments (Figs. 2, 3, 5, 6), the behavioral experiments (Fig. 3), the photoactivation and immunohistochemistry experiments (Figs. 1, 4) have been deposited on the Edmond server, the Open Research Data Repository of the Max Planck Society, which can be accessed via the following link once the article is published: https://doi.org/10.17617/3.88.

The following dataset was generated:

| Author(s) | Year | Dataset title | Dataset URL | Database and Identifier |
|---|---|---|---|---|
| Sudeshna DC | 2022 | Higher-order olfactory neurons in the lateral horn support odor valence and odor identity coding in *Drosophila* | https://edmond.mpdl.mpg.de/dataset.xhtml?persistentId=doi:10.17617/3.88 | doi:10.17617/3.88, 10.17617/3.88 |

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
