## [Editor Report]

Information about the environment, obtained through sensory organs, is processed and utilized at multiple levels in the brain. In this study, the authors use a variety of modern genetic and optophysiological tools to uncover the function and connectivity of glutamatergic neurons in a higher brain center of *Drosophila* – the lateral horn. They find that these neurons do not only encode chemical odor identity, but also the hedonic value (attractive or repulsive) of odors. This advances our understanding of how odors are represented in the brain and will be of value to those who are interested in odor coding and behavioral valence of various odors.

---

## [Decision Letter]

**Decision letter after peer review:**

Thank you for submitting your article "Higher-order olfactory neurons in the lateral horn support odor valence and odor identity coding in *Drosophila*" for consideration by *eLife*. Your article has been reviewed by 3 peer reviewers, including Sonia Sen as Reveiwng Editor and Reviewer #1, and the evaluation has been overseen by K VijayRaghavan as the Senior Editor. The following individual involved in review of your submission has agreed to reveal their identity: André Fiala (Reviewer #2).

Essential revisions:

1. Anatomy:

a. We were concerned about the resolution of the neuroanatomy of the glutamatiergic LHNs as it is presented in this study. Could the authors please improve on this, perhaps by generating single neurons MCFO clones? This could also then be used to compare with the glutamatergic LHNs identified in the Dolan et al., 2019 study.

b. Could the authors please provide a summary of the known connectivity in the LH as a schematic? This will be useful while interpreting the data. We also recommend discussing the odour response patterns observed in this study in the context of this known connectivity.

2. Transection experiments: The authors make the claim that, "…glutamatergic LHNs receive an odour selective inhibition from mPNs which is stronger for the food odour vinegar than for the repellent odour benzaldehyde." This might be too broad a claim to make with a single odour pair, especially given that the mACT control and experiment appear very similar in the benzaldehyde example. Could the authors please repeat this experiment with a second odour pair? We also recommend presenting the control data in the main figure.

3. Stereotypy:

1. Could the authors please demonstrated how reproducible the td-tomato signal is between animals?

2. Could they please show two brains per odour in figure 1A?

In addition to this, could the authors please pay attention to the specific comments below and respond to them wherever possible.

*Reviewer #1 (Recommendations for the authors):*

– It would be nice to have a visual representation of the stereotypy in LHN and uPN responses discussed in figure 2. Could the authors show two brains per odour in figure 1A? I suggest this because as I understand it, the correlation analysis cannot address spatial stereotypy. The linkage analysis suggest it.

– It would be nice to have a better description of the linkage and correlation analyses. I am assuming the each grid was compared across each brain for both these analyses?

– Typo in line 201: It should be 3F, I think.

– The GRASP Experiments: I suggest just retaining the Neurexin results. Do the authors have better images? Perhaps use gray-scale to represent the data better?

– The ablation experiments: I would recommend adding the controls to the main figure. While the iACT ablation results are clear, I'm concerned about the mACT ablation, particularly in light of the controls.

– The Dolan study: Could the authors address their findings in light of this study with respect to neurotransmitter identity, connectivity and behavioural valance. The two approaches are different, and I'm sure there's merit in both.

*Reviewer #3 (Recommendations for the authors):*

I appreciate this study as an important step towards deciphering how odours are represented in the LH and consider this a solid piece of work. I suggest that the presentation and the resolution of the anatomical data should be improved. I also would like to see more than two odours in the experiments defining the iACT versus mACT roles.

1. Presentation of the anatomy data. In general, it will be great to have a representation of LH anatomy that summarizes what is already known about the organization of the LH. Are the response patterns observed for odours of different valence corresponding to those reported in the present study? Eg in Figure 2 it will be helpful to include a 3D representation of the LH, including the imaging planes and the expected volumes of attractive versus aversive odours responses.

2. In Figure 2: how reproducible is the red (td-tomato) signal among animals? This could be a measure of the reproducibility of the positioning of the imaging planes and of the stereotypy of the underlying anatomy.

3. Resolution of the anatomy data. Would it be possible to utilize the EM datasets to improve some of the claims made based on GRASP? Could for instance some of glutamatergic neurons be isolated by MARCM (or by single-cell PA), reconstructed and compared to the existing EM reconstructions? This would help defining for some few exemplary neurons the distribution of pre and postsynaptic sites and the identity of traced partners.

4. Generalization of the data in Figure 5. At line 274: "This observation indicates that the glutamatergic LHNs receive an odour selective inhibition from mPNs which is stronger for the food odour vinegar than for the repellent odour benzaldehyde. " To generalize, could this be tested with other odours? Having done the transection, which seems a very difficult experiment to perform- exposure to an odour panel instead of two odours only should be a relatively easy addition.

5. Would increased odour concentrations (for instance of aversive stimuli) elicit a stronger uPN response and how would that translate into Glutamatergic LHN response amplitudes?

---

## [Author Response]

Essential revisions:1. Anatomy:a. We were concerned about the resolution of the neuroanatomy of the glutamatiergic LHNs as it is presented in this study. Could the authors please improve on this, perhaps by generating single neurons MCFO clones? This could also then be used to compare with the glutamatergic LHNs identified in the Dolan et al., 2019 study.

We agree with the reviewers’ comment regarding the resolution of the neuroanatomy of the glutamatergic LHNs. We have employed photoactivatable GFP to label, trace and reconstruct individual neurons of the glutamatergic subset of LHNs studied here to address this issue. We could successfully label and reconstruct single LHLNs from the dorsomedial cluster (incorporated now into Figure 1C). Due to the position of the cell bodies, the primary neurite as well as the neurotransmitter identity, we assume that these LHLNs belong to the PD3a1 and PD3a2 neuron cluster according to Dolan et al., 2019. This is mentioned in the Results section (page 5).

b. Could the authors please provide a summary of the known connectivity in the LH as a schematic? This will be useful while interpreting the data. We also recommend discussing the odour response patterns observed in this study in the context of this known connectivity.

We have modified the present schematic in Figure 6D according to the reviewers’ suggestion. Along with our data we have also incorporated the connectivity data reported in Dolan et al., 2019, Frechter et al., 2019 and Bates et al., 2020. We have discussed our observed odor-evoked response patterns in glutamatergic LHNs in the context of the known connectivity as suggested in the Discussion part (page13). However, it is rather difficult to predict precisely how the known connectivity based on the EM connectomics data would result in the observed odor-evoked activity patterns in LHNs. First of all, the topographic map of uPN axonal terminals is not retained at the level of LHNs, since the same LHNs receive input from similarly tuned as well as differently tuned uPNs. Secondly, both uPNs as well as mPNs provide input to glutamatergic LHNs and those LHNs in turn feedback onto those second-order neurons and provide feedforward information to other LHNs. Due to this complex network connectivity rather broad than sparse, but odor-specific, activity patterns in LHNs can be expected.

2. Transection experiments: The authors make the claim that, "…glutamatergic LHNs receive an odour selective inhibition from mPNs which is stronger for the food odour vinegar than for the repellent odour benzaldehyde." This might be too broad a claim to make with a single odour pair, especially given that the mACT control and experiment appear very similar in the benzaldehyde example. Could the authors please repeat this experiment with a second odour pair? We also recommend presenting the control data in the main figure.

We agree with the reviewers and according to their suggestion performed the laser transection experiment of mPNs with four other odors, two attractive ones, 2,3 butanedione and ethyl acetate, as well as two aversive odors, acetophenone and linalool. We observed for all odors a trend in increased activity after the laser ablation of mPNs, while only acetophenone displays a significant enhancement in a consistent manner. These observations altogether confirm that glutamatergic LHNs receive an odor-specific inhibition, that is strong for vinegar and acetophenone, moderate for benzaldehyde and weak or absent for 2,3 butanedione, linalool and ethyl acetate. The new data are provided in Figure 5—figure supplement 2 (Results section, page 11). Since Figure 5 fills already one page, it is not possible to include the control data in the main figure (as suggested by the reviewers) without removing data of the transection experiments with benzaldehyde and vinegar and therefore decided to provide the old and new control data in Figure 5—figure supplement 1.

3. Stereotypy:1. Could the authors please demonstrated how reproducible the td-tomato signal is between animals?

Following the reviewers’ suggestion, we provide now representative images of *GH146 mtd-Tomato* across six brains in Figure 2—figure supplement 1A. These images show the high reproducibility of the *GH146 mtd-Tomato* signal and the precise plane that has been used to measure the odor-evoked responses of LHNs across different animals.

2. Could they please show two brains per odour in figure 1A?

According to the reviewers’ recommendation, we added representative images of odor-evoked responses of LHNs and uPNs of two brains in Figure 2—figure supplement 1B and 1C. The images display the spatial stereotypy in odor-evoked responses in both LHNs and PNs between flies.

In addition to this, could the authors please pay attention to the specific comments below and respond to them wherever possible.Reviewer #1 (Recommendations for the authors):– It would be nice to have a visual representation of the stereotypy in LHN and uPN responses discussed in figure 2. Could the authors show two brains per odour in figure 1A? I suggest this because as I understand it, the correlation analysis cannot address spatial stereotypy. The linkage analysis suggest it.

Following the reviewer’s suggestion, we have included representative images of odor-evoked responses of LHNs and uPNs for two brains per odor in Figure 2—figure supplement 1B and 1C. The visual representation confirms that the odor-induced spatial activity patterns are stereotypic across flies in LHNs as well as uPNs.

– It would be nice to have a better description of the linkage and correlation analyses. I am assuming the each grid was compared across each brain for both these analyses?

We agree with the reviewer’s comment and provided a detailed description in the Results (page 6) as well as in the Methods section (page 20). To clarify the reviewer’s doubt, the individual odor-evoked responses of each grid/ pixel were compared only within one animal, not across brains. Therefore, the correlation coefficient and linkage analyses were obtained based on the similarity of responses for all 65 pixels across odors for each animal. The obtained correlation coefficient from each animal was then averaged.

– Typo in line 201: It should be 3F, I think.

I think there is some misunderstanding. The numbering seems correct. 3G shows the correlation between the LHN response and the olfactory preference.

– The GRASP Experiments: I suggest just retaining the Neurexin results. Do the authors have better images? Perhaps use gray-scale to represent the data better?

Although the reviewer suggests keeping only the neurexin GRASP result, we think that it is also informative to show the CD8 GRASP result in parallel. It reveals on one hand the robust signal of CD8 GRASP and on the other hand how the similar pattern of synaptic connectivity is reflected in the neurexin GRASP. Moreover, we would like to keep the colored images, since the green signal in the neurexin GRASP staining clearly shows the sparse synaptic connectivity of glutamatergic LHNs with uPNs and mPNs, which is not so clearly visible in the gray-scaled images.

– The ablation experiments: I would recommend adding the controls to the main figure. While the iACT ablation results are clear, I'm concerned about the mACT ablation, particularly in light of the controls.

We agree with the reviewer’s concern. While repeating the mACT ablation for additional odors, we have also performed the controls for 2,3 butanedione and collected additional data for benzaldehyde, which shows now a lower increase compared to the previous plot. Since Figure 5 fills already one page, it is not possible to include the control data in the main figure (as suggested by the reviewer) without removing data of the transection experiments with benzaldehyde and vinegar and therefore decided to provide the old and new control data in Figure 5—figure supplement 1.

– The Dolan study: Could the authors address their findings in light of this study with respect to neurotransmitter identity, connectivity and behavioural valance. The two approaches are different, and I'm sure there's merit in both.

We agree with the reviewer that both studies have addressed the connectivity of LHNs and their role in determining behavioral valence using different approaches. However, we also think that both studies cannot be simply compared because of their different approaches. Dolan et al., (2019) employed specific split-Gal4 lines to label LHINs, LHLNs and LHONs with different neurotransmitter identities and analyzed their connectivity using EM connectomics. In our study we have used a broad line that labels an ensemble of all glutamatergic LHNs that includes both LHLNs and LHONs and characterized their connectivity using GRASP and functional methods, such as laser transection of specific PN tracts. Dolan et al., employed artificial activation of specific subset of LHNs via CsChrimson to determine their role with regard to attractive and aversive behavioral preferences in diverse behavioral assays. According to them, one class of glutamatergic LHLNs (so-called PV4a1:5) makes excitatory synapses with AV1a1 LHONs which mediate aversive behavioral responses. In our study, since we used a generic Gal4-line to label all glutamatergic LHNs, we could not activate or silence specific neuronal subsets to observe their relevance in odor-guided behavior. Therefore, we correlated the odor response strength of LHNs with the olfactory preference and found that they are tuned to attractive odors and might mediate approach. Since the odor responses in our glutamatergic LHNs includes both LHLNs and LHONs it is fairly difficult to dissect the contribution that each neuron has to determine the behavioral valence. We have added a discussion about the comparison of our study to the data published by Dolan et al., (2019) in the Discussion section (pages 16-17).

Reviewer #3 (Recommendations for the authors):I appreciate this study as an important step towards deciphering how odours are represented in the LH and consider this a solid piece of work. I suggest that the presentation and the resolution of the anatomical data should be improved. I also would like to see more than two odours in the experiments defining the iACT versus mACT roles.1. Presentation of the anatomy data. In general, it will be great to have a representation of LH anatomy that summarizes what is already known about the organization of the LH. Are the response patterns observed for odours of different valence corresponding to those reported in the present study? Eg in Figure 2 it will be helpful to include a 3D representation of the LH, including the imaging planes and the expected volumes of attractive versus aversive odours responses.

We have provided a schematic presentation of connectivity in terms of circuit organization based on previous findings and our study at the end. Providing the schematic only with known findings in the beginning would be rather redundant and may not add to much value.

In one of our previous studies we have observed that the response patterns of mPNs form a valence-specific activation map in the LH (Strutz et al., 2014). Notably, this map corresponds only partly with the response patterns of the glutamatergic LHNs, indicating that the valence code of PNs is not just translated into a valence code at the LH level. We have added this point to our discussion (page 14).

We have measured the odor-evoked responses of glutamatergic LHNs at three different focal planes. However, since the neuronal architecture and synaptic densities are different in different planes (which we have not measured), we think that it would not be precise to extrapolate the functional responses in order to obtain an expected 3D volume of attractive and aversive odors.

2. In Figure 2: how reproducible is the red (td-tomato) signal among animals? This could be a measure of the reproducibility of the positioning of the imaging planes and of the stereotypy of the underlying anatomy.

Following the reviewers’ suggestion, we provide now representative images of *GH146 mtd-Tomato* across six brains in Figure 2—figure supplement 1A. These images show the high reproducibility of the *GH146 mtd-Tomato* signal and the precise plane that has been used to measure the odor-evoked responses of LHNs across different animals.

3. Resolution of the anatomy data. Would it be possible to utilize the EM datasets to improve some of the claims made based on GRASP? Could for instance some of glutamatergic neurons be isolated by MARCM (or by single-cell PA), reconstructed and compared to the existing EM reconstructions? This would help defining for some few exemplary neurons the distribution of pre and postsynaptic sites and the identity of traced partners.

Following the reviewer’s suggestion, we have employed photoactivatable GFP to label, trace and reconstruct individual neurons of the glutamatergic subset of LHNs studied here to address this issue. We have performed single neuron photoactivation of individual cell bodies from the dorsomedial cluster of the glutamatergic LHNs. The neurons have been reconstructed in 3D using the AMIRA software (incorporated now into Figure 1C). Due to the position of the cell bodies, the primary neurite as well as the neurotransmitter identity, we assume that these LHLNs belong to the PD3a1 and PD3a2 neuron cluster according to Dolan et al., 2019. This is mentioned in the Results section (page 5). Unfortunately, Dolan et al., have not discussed potential the pre- or postsynaptic partners of this subset of neurons. Therefore, it is difficult for us to utilize their EM datasets to verify the synaptic partners of our glutamatergic LHNs.

4. Generalization of the data in Figure 5. At line 274: "This observation indicates that the glutamatergic LHNs receive an odour selective inhibition from mPNs which is stronger for the food odour vinegar than for the repellent odour benzaldehyde. " To generalize, could this be tested with other odours? Having done the transection, which seems a very difficult experiment to perform- exposure to an odour panel instead of two odours only should be a relatively easy addition.

We agree with the reviewer and performed the laser transection experiment of mPNs with four other odors, two attractive ones, 2,3 butanedione and ethyl acetate, as well as two aversive odors, acetophenone and linalool. We observed for all odors a trend in increased activity after the laser ablation of mPNs, while only acetophenone displays a significant enhancement in a consistent manner. These observations altogether confirm that glutamatergic LHNs receive an odor-specific inhibition, that is strong for vinegar and acetophenone, moderate for benzaldehyde and weak or absent for 2,3 butanedione, linalool and ethyl acetate. The new data are provided in Figure 5—figure supplement 2 (Results section, page 11). Since Figure 5 fills already one page, it is not possible to include the control data in the main figure (as suggested by the reviewers) without removing data of the transection experiments with benzaldehyde and vinegar and therefore decided to provide the old and new control data in Figure 5—figure supplement 1.

5. Would increased odour concentrations (for instance of aversive stimuli) elicit a stronger uPN response and how would that translate into Glutamatergic LHN response amplitudes?

uPNs activity increases to increased odor concentration in a dose-dependent manner as previously shown. However, in our case the glutamatergic LHN and uPN response amplitudes do not correlate with each other (Figure 3K, Figure 2 Figure Supplement 1B, C) indicating that an increased activity in uPNs does not necessarily evoke an increased activity in LHNs in a linear manner. Hence, it is feasible that a lot of processing is occurring in the LH and that the responses from uPNs are transformed and translated into LHN response patterns in a complex manner. We mention that transformation process in our Discussion.